# COVID-19 patient profiles over four waves in Barcelona metropolitan area: A clustering approach

**Daniel Fernández** [1,2,3] *, **Nuria Perez-Alvarez** [4,5], **Gemma Molist** [6,7], on behalf of the DIVINE project[¶]

**1** Department of Statistics and Operations Research (DEIO), Universitat Politècnica de Catalunya BarcelonaTech (UPC), Barcelona, Spain, **2** Institute of Mathematics of UPC - BarcelonaTech (IMTech), Barcelona, Spain, **3** Centro de Investigación Biomédica en Red de Salud Mental, Instituto de Salud Carlos III (CIBERSAM), Madrid, Spain, **4** Department of Statistics and Operations Research (DEIO), Universitat Politècnica de Catalunya BarcelonaTech (UPC), Barcelona, Spain, **5** Estudis d'Informàtica, Multimèdia i Telecomunicació, Universitat Oberta de Catalunya, Barcelona, Spain, **6** Biostatistics Unit of the Bellvitge Biomedical Research Institute (IDIBELL), L'Hospitalet de Llobregat, Barcelona, Spain, **7** Faculty of Medicine, University of Vic - Central University of Catalonia (UVIC-UCC), Vic, Spain

¶ Membership of the DIVINE project is provided in the Acknowledgments.
* daniel.fernandez.martinez@upc.edu

**Data Availability Statement:** The Ethics Committee of Bellvitge Hospital grants access to data on a study-by-study basis. Additionally, the

## Abstract

### Objectives

Identifying profiles of hospitalized COVID-19 patients and explore their association with different degrees of severity of COVID-19 outcomes (i.e. in-hospital mortality, ICU assistance, and invasive mechanical ventilation). The findings of this study could inform the development of multiple care intervention strategies to improve patient outcomes.

### Methods

Prospective multicentre cohort study during four different waves of COVID-19 from March 1st, 2020 to August 31st, 2021 in four health consortiums within the southern Barcelona metropolitan region. From a starting point of over 292 demographic characteristics, comorbidities, vital signs, severity scores, and clinical analytics at hospital admission, we used both clinical judgment and supervised statistical methods to reduce to the 36 most informative completed covariates according to the disease outcomes for each wave. Patients were then grouped using an unsupervised semiparametric method (KAMILA). Results were interpreted by clinical and statistician team consensus to identify clinically-meaningful patient profiles.

### Results

The analysis included $n_{w1} = 1657$, $n_{w2} = 697$, $n_{w3} = 677$, and $n_{w4} = 787$ hospitalized-COVID-19 patients for each of the four waves. Clustering analysis identified 2 patient profiles for waves 1 and 3, while 3 profiles were determined for waves 2 and 4. Patients allocated in those groups showed a different percentage of disease outcomes (e.g., wave 1: 15.9%

DIVINE research group has agreed that data will only be shared upon request and after evaluation of the purpose and objectives of the study. The DIVINE research group can be contacted via e-mail (grbio@grbio.eu) and the Ethics Committee of Bellvitge Hospital can be also contacted via e-mail (presidenciaceic@bellvitgehospital.cat).

**Funding:** DF, NP, and GM have been supported by l'Agència de Gestió d'Ajuts Universitaris i de Recerca (AGAUR) de la Generalitat de Catalunya (Spain) [2020PANDE00148] (https://agaur.gencat. cat/en/inici/index.html). DF and NP have been supported by the Ministerio de Ciencia e Innovación (Spain) [PID2019-104830RB-I00/ DOI (AEI): 10.13039/501100011033] (https://www.aei. gob.es/en/announcements/announcements-finder/ proyectos-idi-2019-modalidades-retos- investigacion-generacion) and by grant 2021 SGR 01421 (GRBIO, https://agaur.gencat.cat/web/ shared/OVT/Departaments/REU/A_Universitats/ AGAUR/Documents/RECERCA/SGR/Resolucio_ definitiva_SGR-Cat_2021.pdf) administrated by the Departament de Recerca i Universitats de la Generalitat de Catalunya (Spain). The funders had no role in study design, data collection and analysis, decision to publish, or preparation of the manuscript.

**Competing interests:** The authors have declared that no competing interests exist.

(Cluster 1) vs. 31.8% (Cluster 2) for in-hospital mortality rate). The main factors to determine groups were the patient's age and number of obese patients, number of comorbidities, oxygen support requirement, and various severity scores. The last wave is also influenced by the massive incorporation of COVID-19 vaccines.

## Conclusion

Our study suggests that a single care model at hospital admission may not meet the needs of hospitalized-COVID-19 adults. A clustering approach appears to be appropriate for helping physicians to differentiate patients and, thus, apply multiple care intervention strategies, as another way of responding to new outbreaks of this or future diseases.

## 1. Introduction

Since the first case of respiratory syndrome coronavirus (SARS-CoV-2, also known as COVID-19) in Wuhan [1], the disease has spread rapidly around the world. The World Health Organization (WHO) declared a global pandemic in March 2020. In Catalonia (Spain), the first detected case was in February 2020, and there have been six declared waves since then, with 1.3 million cases and more than 24 thousand deaths [2].

The entire population is generally susceptible to the virus and the symptoms of COVID-19 are diverse as they can be from asymptomatic patients to severe pneumonia or even death. Thus, there has been a lot of interest in trying to find the COVID-19 patient's profiles and their associated risk factors to determine the worst prognosis using as endpoints admission to intensive care unit (ICU), the requirement of invasive mechanical intubation (IVM), and death [3].

Clustering techniques are one of the most well-known unsupervised learning methods. They have been successfully applied in medical research [4–6] and allow us to reveal diverse group profiles and their patterns of association among factors [7]. A clustering approach can help us to both predict the outcome of COVID-19 in patients and to reveal the set of key variables, which clinicians could consider when evaluating patients and making early decisions for future outbreaks [8].

There has been research focused on a single outcome of COVID-19 and the prevalence of specific symptoms [9–11]. The most studied characteristics as risk factors are age, comorbidities, and clinical analytics [12–16]. Additionally, research using prediction models for diagnosis and prognosis is common [17, 18]. To the best of our knowledge, determining patient profiles involving the combination of relevant comorbidities, clinical results, and sociodemographic features and their relationship with relevant disease outcomes such as the requirement of IVM, ICU assistance, and in-hospital mortality has been less studied [8, 19–23].

We hypothesize that hospitalized-COVID-19 patients can be comprised into groups with different characteristics associated with clinical outcomes. Thus, our research aims to study those characteristics for hospitalized-COVID-19 patients in a Barcelona metropolitan region via a clustering method to identify specific patient profiles across four waves and their association with relevant outcomes of disease.

## 2. Methods

### a. Setting and participants

We performed a prospective multicentre cohort study during four waves of COVID-19 (the first wave included hospitalized-COVID-19 patients between March 1st and April 15th, 2020;

the second wave, from October 1[st] to November 30[th], 2020; the third, from January 1[st] to February 28[th], 2021; and the fourth, from July 1[st] to August 31[st], 2021) in four health consortiums (Hospital Universitari de Bellvitge, Consorci Sanitari de l'Alt Penedès I Garraf, Hospital de Viladecans, and Hospital de Sant Boi de Llobregat) located in the Barcelona metropolitan south region (eFig 1 in S1 File depicts that geographical region), covering a population of 1,370,709 inhabitants (eTable1 of the S1 File shows the number of patients included in the study broken down by health consortiums and number of wave). All patients were adults (>18 years old) and were admitted with a PCR-proven SARS-CoV-2 infection that occurred before admission, with a maximum of 48 hours between the positive test and admission.

The study was approved by the ethics committee (CEIm Hospital Universitari de Bellvitge) in accordance with Spanish legislation and was performed in accordance with the Helsinki Declaration of 1964. The need for patient informed consent was waived by the ethics committee.

## b. Analytic process

Our aim was to identify distinct hospitalized-COVID-19 patients groups with different characteristics associated with clinical outcomes via the application of a 2-step procedure: 1) a supervised method (classification tree) to select the most relevant factors, which avoids redundancy, and 2) an unsupervised semiparametric method (KAMILA) [24], kAy-means for mIxed lArge data) for clustering mixed-type data. We implemented the following four steps to achieve this goal:

**1) Collecting and formatting of study data.** Study data were collected and managed using REDCap [25, 26]. The questionnaires were designed by joint agreement of the clinical and statistician teams and filled in by physicians in each hospital. For each wave, the collected information for each hospitalized-COVID-19 patient is essentially the same and includes a diverse range of features at admission time as demographic characteristics (e.g. gender and age), the prevalence of comorbidities (e.g., smoke and Charlson index), previous treatments (e.g., statins and corticosteroids), infection symptoms and clinical exploration (e.g., oxygen saturation, respiratory frequency, and temperature), analytics (e.g., leucocytes and neutrophils) and severity scores (pneumonia severity index, pneumonia severity score (CURB-65), and viral pneumonia mortality score). We also note that the last wave (from July 1[s]t to August 3[1s]t, 2021) includes the information about vaccination (e.g., vaccination dose received: none, partial, or full regime). Additionally, meaningful outcomes of COVID-19 such as the requirement of IVM, ICU assistance, and in-hospital mortality were also recorded during the admission period time. Finally, information about patients' ceiling of care was included [27], which was defined as the maximum therapeutic effort to be offered to a patient based on their age, their associated comorbidities, and the expected clinical benefit concerning the availability of resources.

All data was merged including an indicator variable of wave and, after the data collection was completed, an exhaustive data recovery process was performed to avoid missing information and to overcome possible capture errors during the data entry period.

**2) Guided variable selection and reduction.** Decisions about variables were made by clinical and statistician team consensus. Variables were dichotomized using common clinical thresholds. From the initial set of 292 variables, we identified the subset of variables that had the most relative influence to discriminate concerning the disease outcomes for each wave by using CART [28]. The results were also validated with a logistic and Cox stepwise regression with bootstrapping, being all results consistent. From that selection, we ranked the variables by the percentage of missingness to determine the definitive set of variables used to identify clusters. Thus, we used the completed 36 variables listed in eTable 2 of the S1 File. All 36 variables were utilized in the determination of patients' clusters. However, Tables 1 and 2 only display

the 19 statistically significant variables, along with gender for demographic interest, resulting in a total of 20 variables.

**3) Determination of patients' clusters.** We applied the KAMILA (KAy-means for MIxed LArge data) method, a semiparametric clustering approach for mixed-type data, which is able to combine the contribution of continuous and categorical variables without strong parametric assumptions. Several studies have demonstrated KAMILA's superiority in handling high imbalances between continuous and categorical data compared to other methods [29]. The KAMILA algorithm is a development of the k-means method who combines Gaussian-multinomial mixture models [30], a model-based approach, with the k-means algorithm [31, 32], a non-model-based approach, both of which have been adapted successfully to very large data sets [33, 34]. In the KAMILA procedure, each parameter is initialized by randomly drawing

**Table 1. Comparison of demographic characteristics, comorbidities, vital signs and severity scores at hospital admission between clusters for wave 1 (March 1st—April 15th, 2020).** The table only shows the statistically significant variables[1].

| Wave 1 (2 clusters) | Cluster 1 ($N_{w1}$ = 1226, 74.0%) | Cluster 2 ($N_{w1}$ = 431, 26.0%) |
|---|---|---|
| *Demographic characteristics and comorbidities* | | |
| **Age**, Median [Q1; Q3] | 65.0 [53.0,76.0] | 70.0 [57.5,79.0] |
| **Gender—Women**, N (%) | 499 (40.7) | 159 (36.9) |
| **Obesity (BMI > 30)**, N (%) | 367 (35.4) | 139 (40.1) |
| **Charlson Index**, Mean (SD) | 3.2 (2.6) | 3.7 (2.4) |
| **Dementia**, N (%) | 60 (4.9) | 39 (9.0) |
| **COPD**, N (%) | 217 (17.7) | 98 (22.7) |
| **Heart failure**, N (%) | 79 (6.4) | 42 (9.7) |
| **Hypertension**, N (%) | 594 (48.5) | 249 (57.8) |
| **Degenerative neurological disease**, N (%) | 24 (2.0) | 18 (4.2) |
| **Peripheral vascular disease**, N (%) | 38 (3.1) | 25 (5.8) |
| *Vital signs, severity scores, and clinical analytics at hospital admission* | | |
| **FiO$_2$**, Mean (SD) | 21.0 (1.4) | 48.8 (18.3) |
| **SatO$_2$/FiO$_2$**, Mean (SD) | 447.0 (32.0) | 254.0 (118.0) |
| **ROX index**, Mean (SD) | 22.3 (5.8) | 11.7 (7.0) |
| **PSI group**, N (%) | | |
| 1 | 647 (52.8) | 134 (31.1) |
| 2 | 250 (20.4) | 111 (25.7) |
| 3 | 253 (20.6) | 130 (30.2) |
| 4 | 76 (6.2) | 56 (13.0) |
| **CURB-65 group**, N (%) | | |
| Low risk | 842 (68.7) | 237 (55.0) |
| Intermediate risk | 267 (21.8) | 108 (25.0) |
| High risk | 117 (9.5) | 86 (20.0) |
| **MuLBSTA**, Mean (SD) | 7.9 (3.5) | 8.9 (3.3) |
| **Severe Pneumonia**, N (%) | 407 (33.2) | 267 (61.9) |
| **Respiratory rate**, Mean (SD) | 21.5 (6.1) | 24.4 (7.3) |
| **Lymphosits (mil/mm$^3$)**, Mean (SD) | 1.1 (1.4) | 1.0 (1.2) |
| **Angiotensin receptor antagonists**, N (%) | 160 (13.1) | 81 (18.8) |

[1]. Based on Mann-Whitney U test for numerical variables and Chi-squared tests for categorical variables. Gender was not significant, but it was included for demographic interest;

[2]. BMI: Body Mass Index, PSI: Pneumonia Severity Index, COPD: Chronic pulmonary pathology, ROX: Ratio of oxygen saturation, MuLBSTA: Viral Pneumonia Mortality Score; CURB-65: Pneumonia severity score [47]

**Table 2. Comparison of demographic characteristics, comorbidities, vital signs and severity scores at hospital admission between clusters for wave 4 (July 1st—August 31st, 2021).** The table only shows the statistically significant variables[1].

| Wave 4 (3 clusters) | Cluster 1 ($n_{w4}$ = 346, 44.0%) | Cluster 2 ($n_{w4}$ = 205, 26.0%) | Cluster 3 ($n_{w4}$ = 236, 30.0%) |
|---|---|---|---|
| *Demographic characteristics and comorbidities* | | | |
| **Age**, Median [Q1; Q3] | 43.0 [34.2,55.0] | 52.0 [39.0,65.0] | 78.5 [67.0,86.0] |
| **Gender—Women**, N (%) | 121 (35.0) | 83 (40.5) | 89 (37.7) |
| **Obesity (BMI > 30)**, N (%) | 95 (27.5) | 94 (45.9) | 86 (36.4) |
| **Charlson Index**, Mean (SD) | 1.1 (1.6) | 2.0 (2.2) | 5.0 (2.3) |
| **Dementia**, N (%) | 1 (0.3) | 3 (1.5) | 36 (15.3) |
| **Heart failure**, N (%) | 4 (1.2) | 11 (5.4) | 13 (18.2) |
| **Hypertension**, N (%) | 18 (5.2) | 65 (31.7) | 198 (83.9) |
| **Vaccinated (partial or full)**, N (%) | 89 (25.7) | 45 (22.0) | 217 (91.9) |
| **Vaccination dose received**, N (%) | | | |
| No vaccine | 257 (74.3) | 160 (78.0) | 19 (8.1) |
| Partial regimen | 36 (10.4) | 11 (5.4) | 12 (5.1) |
| Full regimen | 53 (15.3) | 34 (16.6) | 205 (86.8) |
| *Vital signs, severity scores, and clinical analytics at hospital admission* | | | |
| **FiO$_2$**, Mean (SD) | 23.0 (4.4) | 54.2 (28.1) | 26.7 (9.4) |
| **SatO$_2$/FiO$_2$**, Mean (SD) | 427.0 (54.0) | 233.0 (84.0) | 384.0 (83.6) |
| **SatO$_2$**, Mean (SD) | 96.0 (2.2) | 94.1 (4.8) | 95.4 (2.8) |
| **ROX index**, Mean (SD) | 21.9 (4.8) | 11.3 (6.9) | 19.9 (6.3) |
| **PSI group**, N (%) | | | |
| 1 | 279 (80.6) | 109 (53.2) | 26 (11.0) |
| 2 | 40 (11.6) | 48 (23.4) | 47 (19.9) |
| 3 | 27 (7.8) | 39 (19.0) | 113 (47.9) |
| 4 | 0 (0.0) | 9 (4.4) | 50 (21.2) |
| **CURB-65 group**, N (%) | | | |
| Low risk | 323 (93.3) | 148 (72.2) | 92 (39.0) |
| Intermediate risk | 21 (6.1) | 49 (23.9) | 97 (41.1) |
| High risk | 2 (0.6) | 8 (3.9) | 47 (19.9) |
| **MuLBSTA**, Mean (SD) | 6.4 (2.9) | 8.8 (3.2) | 9.2 (3.5) |
| **Respiratory rate**, Mean (SD) | 20.2 (3.9) | 22.5 (6.3) | 20.3 (4.7) |
| **Neutrophils (mil/mm$^3$)**, Mean (SD) | 4.8 (3.0) | 7.0 (4.1) | 5.9 (3.4) |
| **Lymphosits (mil/mm$^3$)**, Mean (SD) | 1.1 (0.5) | 0.8 (0.4) | 1.3 (2.2) |
| **Angiotensin receptor antagonists**, N (%) | 11 (3.2) | 19 (9.3) | 55 (23.3) |
| **Angiotensin-converting enzyme**, N (%) | 2 (0.6) | 30 (14.6) | 92 (39.0) |

[1]. Based on Mann-Whitney U test for numerical variables and Chi-squared tests for categorical variables. Gender was not significant, but it was included for demographic interest;

[2]. BMI: Body Mass Index, PSI: Pneumonia Severity Index, COPD: Chronic pulmonary pathology, ROX: Ratio of oxygen saturation, MuLBSTA: Viral Pneumonia Mortality Score; CURB-65: Pneumonia severity score [47]

from a uniform distribution with bounds set to the minimum and maximum of each continuous variable. Categorical variables are initialized by drawing from a Dirichlet distribution with shape parameters all equal to one. The algorithm is initialized multiple times. For each initialization, the algorithm iterates until either reaching a pre-specified maximum number of iterations or until the population membership remains unchanged from the previous iteration, whichever occurs first. In our study, the number of clusters was determined by analyzing 2–10

group models. KAMILA has been applied in recent health studies [35–38] to stratify individuals into clusters according to the outcomes of interest.

**4) Interpretation of cluster profiles.** We used a 3-fold strategy to derive clinical and sociodemographic meaning from the resulting clustering structures. Firstly, we compared the results of the patients' groups for the first wave obtained with KAMILA with those obtained applying the k-means algorithm [39],which uses only numerical variables. Secondly, we compare the groups among the four COVID-19 waves to investigate patterns of commonality and dissimilarity. Thirdly, we presented the clustering structures resulting from the statistical analysis to physicians who are part of the DIVINE research team and worked with them to define clinically meaningful patient profiles. During this procedure, we discussed the medical interpretation and contextualization of the results, and reached a consensus on the definitions.

## c. Statistical analysis

All statistical analysis and graph display was carried out using the statistical software R version 4.0.2 (R Project for Statistical Computing).

**Classification trees and bootstrapping.** The assessment of the importance of the variables according to the relevant dichotomous relevant disease outcomes (in-hospital mortality, ICU assistance, and invasive mechanical ventilation) was determined using a classification tree methodology (CART [28]) based on Gini's impurity index as the splitting criterion. CART was implemented with the R package `rpart` [40].

**KAMILA.** The application of the KAMILA method was carried out using the `kamila` [41] R package. Determining the clustering tendency of the data (i.e., its clusterability) was assessed using Hopkins' statistic [42]. Model selection to decide the optimal number of clusters was performed according to the prediction strength method [43]. The prediction strength indicates the proportion of pairs of cases classified in the same cluster in both the training and test samples, considering the cluster with the worst performance. Since a separate test sample is often not available, the procedure employs repeated two-fold cross-validation. In this process, the first fold serves as the training sample, while the second fold serves as the test sample. Moreover, it is important to remark that class sample sizes were also considered since inadequate sample sizes can lead to convergence problems.

**Post-hoc analysis.** For each clustering structure, the differences of the categorical variables were measured using the Pearson's Chi-squared test and continuous variables were compared using the Mann-Whitney U test or Kruskal-Wallis test (including pairwise comparisons) to assess differences between clusters for two or more than two cluster solutions, respectively. A two-sided p-value $< 0.05$ was considered statistically significant. Finally, result among waves has been compared for consistency.

# 3. Results

## Patient population

We included $n_{w1} = 1657$, $n_{w2} = 697$, $n_{w3} = 677$, and $n_{w4} = 787$ hospitalized-COVID-19 patients for each of the four waves analyzed. Their Hopkins' statistic measures range between 0.82 to 0.94, which indicates that they are highly clustered. The median (IQR) age of the samples was 66 [54, 76], 65 [55, 77], 66 [54, 77], and 56 [40, 73] years old (y.o.), women range 36.7%–42.7%, Charlson comorbidity index [44] ranges between 2.5 and 3.5, and percentage of oxygen saturation between 93.6% and 95.3%. The in-hospital mortality, ICU assistance, and invasive mechanical ventilation rates ranges among the four waves between [10.2%,20.0%], [12.0%,15.8%] and [8.3%,10.8%], respectively. A more comprehensive description of all variables can be seen in eTable 2 of the S1 File.

## KAMILA results

The optimal number of patient groups identified was two for waves 1 and 3 and three for waves 2 and 4. The prediction strength values for the KAMILA clustering in the four waves are depicted in eFig 2 of the S1 File. Generally speaking, the variables that were found to be most important in determining the clusters for the first three waves included the patient's age, the number of obese patients, the number of comorbidities, the requirement for oxygen support, and various severity scores such as SaFi illness [45, 46]. SaFi illness is defined as the ratio between the oxygen saturation and the fraction of inspired oxygen. (i.e., $SatO_2/FiO_2$). For the last wave, the vaccination dose that the patient has received is the most determining variable.

## Patient profiles

Tables 1 and 2 show the resulting clustering structures for waves 1 and 4, respectively (see eTables 3 and 4 of the S1 File for waves 2 and 3, respectively) and eFig 4 in S1 File depicts a graphical comparison of sociodemographic and clinical characteristics[1] between KAMILA clusters for each wave. In wave 1 (Table 1), 74.0% of patients ($n_{w1}$ = 1226) were grouped into Cluster 1, which allocates patients with the best prognostic (see Fig 1 and Table 3). Those patients are characterized by lower ages (median [IQR]: 65[53–76] (Cluster 1) vs. 70[57–79] (Cluster 2)), smaller and not dispersed $FiO_2$ value, larger $SatO_2/FiO_2$ values, and consistently, a lower number of age-related comorbidities and scores in pneumonia-related measures (PSI, CURB-65, and MuLBSTA). On the other hand, the set of patients assigned to Cluster 2 presents more aging-related diseases such as dementia, chronic pulmonary pathology, heart failure, hypertension, degenerative neurological disease, and peripheral vascular diseases. It also includes more men and a greater number of obese patients.

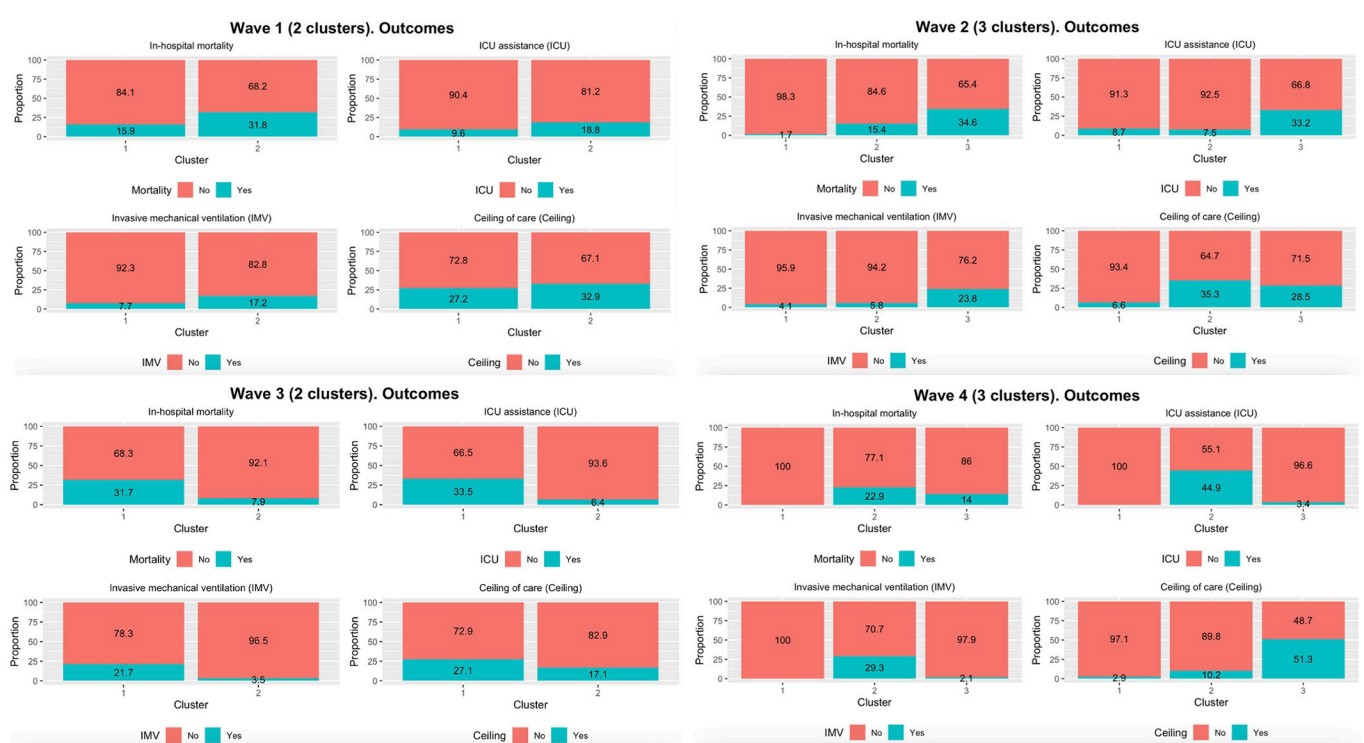

**Fig 1. Bar plots of the % of the outcome (in-hospital mortality, ICU assistance, and invasive mechanical ventilation) and % of ceiling of care broken down by cluster.**

**Table 3. Outcomes (in-hospital mortality, ICU assistance, and invasive mechanical ventilation) according to patient profiles defined by clustering in each wave.** Sample sizes and proportion for each cluster are indicated, frequencies are shown in each cell and the figures between parenthesis are the % of the outcome for that determined cluster.

| Wave 1 (2 clusters) Outcomes | Cluster 1 ($n_{w1}$ = 1226, 74.0%) | | Cluster 2 ($n_{w1}$ = 431, 26.0%) | p-value |
|---|---|---|---|---|
| In-hospital mortality | 195 (15.9%) | | 137 (31.8%) | <0.001 |
| ICU assistance | 118 (9.6%) | | 81 (18.8%) | <0.001 |
| Invasive mechanical ventilation | 95 (7.7%) | | 74 (17.2%) | <0.001 |
| **Wave 2 (3 clusters) Outcomes** | **Cluster 1 ($n_{w2}$ = 242)** | **Cluster 2 ($n_{w2}$ = 241)** | **Cluster 3 ($n_{w2}$ = 214)** | **p-value** |
| In-hospital mortality | 4 (1.7%) | 37 (15.4%) | 74 (34.6%) | <0.001 |
| ICU assistance | 21 (8.7%) | 18 (7.5%) | 71 (33.2%) | <0.001 |
| Invasive mechanical ventilation | 10 (4.1%) | 14 (5.8%) | 51 (23.8%) | <0.001 |
| **Wave 3 (2 clusters) Outcomes** | **Cluster 1 ($n_{w3}$ = 221)** | | **Cluster 2 ($n_{w3}$ = 456)** | **p-value** |
| In-hospital mortality | 70 (31.7%) | | 36 (7.9%) | <0.001 |
| ICU assistance | 74 (33.5%) | | 29 (6.4%) | <0.001 |
| Invasive mechanical ventilation | 48 (21.7%) | | 16 (3.5%) | <0.001 |
| **Wave 4 (3 clusters) Outcomes** | **Cluster 1 ($n_{w4}$ = 346)** | **Cluster 2 ($n_{w4}$ = 205)** | **Cluster 3 ($n_{w4}$ = 236)** | **p-value** |
| In-hospital mortality | 0 (0.0%) | 47 (22.9%) | 33 (14.0%) | <0.001 |
| ICU assistance | 0 (0.0%) | 92 (44.9%) | 8 (3.4%) | <0.001 |
| Invasive mechanical ventilation | 0 (0.0%) | 60 (29.3%) | 5 (2.1%) | <0.001 |

Note: p-value is from Pearson's Chi-squared test with Yates' continuity correction

In Wave 2 (see eTable 3 in S1 File), the patients were divided into three clusters. Cluster 3 had a higher percentage of men, more obese patients, and presented worse prognostic indicators (see Fig 1). Cluster 1 included younger patients with fewer comorbidities, while the opposite was found for patients in Cluster 2.

For patients in Wave 3 (see eTable 4 in S1 File), two clusters were identified. There were no significant differences in the number of comorbidities (Charlson index) between the clusters, but Cluster 1 contained patients with worse health outcomes, lower $SatO_2/FiO_2$ levels, older age, higher pneumonia scores, and more severe pneumonia symptoms.

In wave 4 (Table 2), we observe a clear vaccination effect in the classification of the patients among clusters. Cluster 3 has allocated the elderly patients (median [IQR]: 78.5 [67, 86]), who were fully vaccinated (86.8%) in comparison with the other two clusters of patients (Clusters 1 and 2 present the larger percentage of non-vaccinated patients: 74.3% and 78.0%, respectively). Between the two less-vaccinated groups of patients, those allocated in Cluster 2 presented worse health outcomes, a small ratio of full regimen vaccination (16.6%) combined with a wider age's range (median [IQR]: 52 [39, 65]). Furthermore, patients in Cluster 2 presented a smaller $SatO_2/FiO_2$ (mean (SD): 233.0 (84.0)) and worse viral pneumonia mortality score (MuLBSTA) than in patients in Cluster 1. The elderly patients (Cluster 3), despite having a larger number of comorbidities, are not labeled as the worst prognostic group as they received full-regimen vaccine and could have potentially survived the previous waves. In the current wave (July 1st—August 31st, 2021), they may have also received full-regimen vaccination, which could have contributed to their improved prognosis.

A visualization of data of all profiles described here via the first two first components from Principal Components Analysis is shown in eFig 3 of the S1 File.

The characteristics highlighted in the identified clusters of patients lead to different degrees of disease severities of COVID-19 outcomes (variables not used to determine the clusters), which were compared across the clusters in each wave (Fig 1). Differences in the outcomes

IVM, ICU assistance, and in-hospital mortality are obvious over all waves (see Table 3 for statistical tests). Thus, the profiles of the patients with a worse prognosis in the COVID-19 outcomes are those that correspond to the most severe clinical characteristics.

## 4. Discussion

This study identified grouping structures of two (waves 1 and 3) and three (waves 2 and four) hospitalized-COVID-19 patient profiles across the four waves using the clinical and demographic information and applying a clustering approach, which allowed data learning without any prior hypothesis. In our view, patients' profiles can be clearly divided between waves 1–3 and wave 4. Some profiles were narrowly close in their descriptions between the first three waves. Thus, the clustering approach always determined a group of hospitalized patients who had the worse rates of COVID-19 outcomes and were characterized to be elderly obese men with a medium/large number of comorbidities and low degree of vital signs and severity scores. Generally speaking, the other clusters found in these three first waves are distinguished by the number of comorbidities and age. The number of vaccination doses received is crucial to discriminate groups in wave 4: Clusters 1–2 had a 22–25% of partial or full vaccinated rate vs. the 92% in Cluster 3. Moreover, age is probably the most important variable to differentiate between the profiles of the first two clusters: Cluster 1 (median [IQR]: 43[34, 55]) and 2 (median [IQR]: 52[39, 65]) as the percentage of full-regimen vaccination was similar (15–16%). As explained above in the results section, our hypothesis is that Cluster 3 in wave 4 are elderly people who potentially survived across the first three waves.

We observed that the COVID-19 patient profiles we identified exhibited varying degrees of COVID-19 outcomes, including in-hospital mortality, ICU assistance, and invasive mechanical ventilation. Interestingly, these outcomes were not used in the process of determining the clusters. We believe this may be because the selection of relevant variables to use in the clustering process was performed using a supervised method based on those COVID-19 outcomes. Thus, this finding has significant implications as can provide insight into the patient's potential prognostic presentation based on the observation of the clinical and socio-demographic characteristics provided at baseline. Additionally, although the patient's ceiling of care is a variable that must be taken with caution since it can change between waves and centers, Fig 1 shows that patients with the ceiling of care had poorer in-hospital mortality than patients without ceiling of care. For a more extensive discussion about this topic, we suggest reading Pallarés *et al.* [27]. which deals with the same group of patients.

There were differences in treatments received during hospital admission across waves. For example, treatments used in the first wave, such as hydroxychloroquine and lopinavir/ritonavir, were found to be ineffective and were replaced by others like remdesivir and corticosteroids. This is observed in the improvement of COVID-19 outcomes. For instance, we can see in Table 3 that the percentages of in-hospital mortality are 20.0% (wave 1), 16.5% (wave 2), 15.6% (wave 3), and 10.6% (wave 4). The last wave is also influenced by the massive incorporation of COVID-19 vaccines. This derived in that the clustering methodology for wave 4 determined a group of 346 patients (Table 2—Cluster 1, 44% of the total of the sample) with no in-hospital mortality, ICU, or IVM (Fig 1).

This study emphasized the heterogeneity among hospitalized-COVID-19 patients and how it could be associated to different degrees of severity of the studied disease outcomes. Some of the strengths of this study are its multicenter framework which provided a large number of patients covering a region and the inclusion of a vast number of demographic and clinical features. However, several limitations must be remarked: 1) This research did not intend to explain causality, only statistical associations and descriptions. Our results are obtained using

the extent of observational data available. Ideally, we should have determined the clusters in a randomized design, but such design is impossible to pursue in the current settings; 2) the study cohort consisted exclusively of COVID-19-hospitalized patients in the Barcelona metropolitan south region and did not include outpatients, which can cause a selection bias. Thus, our results cannot be generalized to the general population; 3) our clustering approach focused on completed data. To address this limitation, we statistically compared the profiles of completed record patients with those non-completed. We obtain equivalent clinical and sociodemographic profiles. However, handling missingness, for instance using a multiple imputation strategy, is a possible and robust strategy to follow in future studies; 4) In contrast to predictive modelling, there are currently no unique gold standards for statistical validation of data clustering results. To overcome this restriction, we assessed validity within three domains [19]: face validity (the clustering structure obtained in each wave was recognizable by the clinical team and was reasonable and coherent given their expertise), construct validity (the resulting clustering structures using two unrelated clustering approaches (k-means and KAMILA) are equivalent), and criterion validity (the different patient groups has significantly different outcomes of disease); 5) the applied methodologies in this study may not capture fully the trends and patterns of the evolving COVID-19 pandemic, since are time-limited; future research over a longer period of time would be needed.

Further research should also consider determining the profiles stratifying by the ceiling of care information or adding information about outpatients to avoid selection bias. Additionally, our last general consideration would be that statistical methods such as the ones performed here can potentially exhibit hidden patterns in the data, but the consensus of the results with clinical teams is imperative to give clinical context and sensible understanding. We believe that our research has the potential to serve as a guideline for epidemiologists worldwide, extending its applicability beyond Spain.

## 5. Conclusions

Our study suggests that a single care model at hospital admission may not meet the needs of hospitalized-COVID-19 adults according to age, obesity, oxygen vital signs, high comorbidity, and severe pneumonia scores. A clustering approach appears appropriate to help physicians differentiate patients based on their profile and possible disease outcomes and, thus, apply multiple care intervention strategies, as another way of responding to new outbreaks of this or future diseases.

## Supporting information

**S1 File.**
(DOCX)

## Acknowledgments

*Divine Study group*

The DIVINE research team is composed by (apart from the three authors of this article): Gabriela Abelenda Alonso[1,2]; Mireia Besalú[2]; Jordi Carratalà[1,3,4,5]; Erik Cobo[6]; Jordi Cortés[6]; Leire Garmendia[6]; Guadalupe Gómez-Melis[6]; Carlota Gudiol[1]; Pilar Heureu[1], Natàlia Pallarès[7]; Klaus Langohr[6]; Xavier Piulachs[6]; Alexander Rombauts[1,2]; Cristian Tebé[7]; Sebastián Videla[8,9]

[1]Department of Infectious Diseases, Bellvitge University Hospital, Barcelona, Spain.
[2]Bellvitge Biomedical Research Institute (IDIBELL), Barcelona, Spain.

[3]Department of Clinical Sciences, School of Medicine and Health Sciences, University of Barcelona, Barcelona, Spain.

[4]Bellvitge Biomedical Research Institute (IDIBELL), Barcelona, Spain.

[5]CIBERINFEC, Instituto de Salud Carlos III, Madrid, Spain.

[6]Department of Statistics and Operations Research, Universitat Politècnica de Catalunya/ Barcelonatech, Barcelona, Spain.

[7]Biostatistics Unit of the Bellvitge Biomedical Research Institute (IDIBELL), L'Hospitalet de Llobregat, Avinguda de la Granvia de l'Hospitalet, 199, 08908, Barcelona, Spain.

[8]Department of Clinical Pharmacology, Bellvitge University Hospital, Barcelona, Spain.

[9]Department of Pathology and Experimental Therapeutics, School of Medicine and Health Sciences, University of Barcelona, Barcelona, Spain.

The leaders (and e-mail addresses) of this group are Cristian Tebé (ctebe@idibell.cat) and Guadalupe Gómez-Melis (lupe.gomez@upc.edu).

## Author Contributions

**Conceptualization:** Daniel Fernández, Nuria Perez-Alvarez, Gemma Molist.

**Data curation:** Gemma Molist.

**Formal analysis:** Daniel Fernández, Nuria Perez-Alvarez.

**Funding acquisition:** Daniel Fernández, Nuria Perez-Alvarez.

**Investigation:** Daniel Fernández, Nuria Perez-Alvarez.

**Methodology:** Daniel Fernández, Nuria Perez-Alvarez.

**Resources:** Daniel Fernández, Nuria Perez-Alvarez.

**Software:** Gemma Molist.

**Supervision:** Daniel Fernández.

**Validation:** Daniel Fernández.

**Visualization:** Daniel Fernández, Nuria Perez-Alvarez, Gemma Molist.

**Writing – original draft:** Daniel Fernández.

**Writing – review & editing:** Daniel Fernández, Nuria Perez-Alvarez, Gemma Molist.

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
