## [Author Response · Author response to Decision Letter 0]

8 May 2023

We have included a rebuttal letter that responds to each point raised by the academic editor and reviewers (file: Response_to_Reviewers.docx), with this information:

Sonu Bhaskar, MD PhD

Academic Editor

PLOS ONE

April 21, 2023

Dear Drs. Bhaskar,

We are very grateful for the opportunity to revise our manuscript “COVID-19 patient profiles over four waves in Barcelona metropolitan area: A clustering approach” (Manuscript Number: PONE-D-22-31704) for PLOS ONE. 

We carefully considered the reviewers' comments, and we explained how we have revised the manuscript based on those comments and recommendations. Furthermore, we hope that those revisions improve the paper such that you and the reviewers now deem it worthy of publication in PLOS ONE. We submitted detailed responses to the reviewers' comments (written in blue). 

For the revised manuscript, all changes are tracked to ease their identification.

We have included the following items when submitting your revised manuscript:

• Response_to_Reviewers.docx: A rebuttal letter that responds to each point raised by the academic editor and reviewers. 

• Profiles_Revised_Manuscript_with_TrackChanges.docx: A marked-up copy of our manuscript that highlights changes made to the original version. 

• Profiles_Revised_Manuscript.docx: An unmarked version of our revised paper without tracked changes. 

Our revised paper focused on the following key points:

• We amended the description of the funding information and the data availability, following the Journal Requirements listed by the Academic Editor.

• We rewrote quite a few parts of the article following reviewers comment and suggestions to make it more clear. 

• We moved one table (current Table 3) from supplementary material to the main body, which makes the manuscript more comprehensive.

• We added a better quality image of Figure 1.

• We extended the state-of-art, adding three references.

• We thoroughly proofread the whole manuscript. 

Additionally, as required in the Journal Requirements listed by the Academic Editor (Dr. Sonu Bhaskar), we removed the funding information from the revised version of the manuscript and included the following amended statements here (we understand you will change it in the online submission form on our behalf):

DF, NP, and GM have been supported by l’Agència de Gestió d'Ajuts Universitaris i de Recerca (AGAUR) de la Generalitat de Catalunya (Spain) [2020PANDE00148] (https://agaur.gencat.cat/en/inici/index.html). DF and NP have been supported by the Ministerio de Ciencia e Innovación (Spain) [PID2019-104830RB-I00/ DOI (AEI): 10.13039/501100011033] (https://www.aei.gob.es/en/announcements/announcements-finder/proyectos-idi-2019-modalidades-retos-investigacion-generacion) and by grant 2021 SGR 01421 (GRBIO) administrated by the Departament de Recerca i Universitats de la Generalitat de Catalunya (Spain) (https://tauler.seu.cat/pagDetall.do?idEdicte=419801&idens=1). The funders had no role in study design, data collection and analysis, decision to publish, or preparation of the manuscript.

Regarding the Data Availability statement (we understand you will change it in the online submission form on our behalf): 

The Ethics Committee of Bellvitge Hospital grants access to data on a study-by-study basis. Additionally, the DIVINE research group has agreed that data will only be shared upon request and after evaluation of the purpose and objectives of the study. The corresponding author of the manuscript can be contacted, who can address the request to the Ethics Committee of Bellvitge Hospital and DIVINE research group.

We confirm that this manuscript describes an original work and has not been published or under consideration elsewhere, and is not under consideration by another journal. All authors have approved the manuscript and agree with submission to PLOS ONE. The authors have no conflicts of interest to declare, and all of them fulfill the criteria to be considered authors due to their contributions.

We hope that the editorial board will agree on the interest of this study and look forward to hearing from you at your earliest convenience.

Yours sincerely, 

Daniel Fernández on behalf of the authors

---

## [Decision Letter · Decision Letter 1]

8 Mar 2024

PONE-D-22-31704R1COVID-19 patient profiles over four waves in Barcelona metropolitan area: A clustering approachPLOS ONE

Dear Dr. Fernández,

Thank you for submitting your manuscript to PLOS ONE. After careful consideration, we feel that it has merit but does not fully meet PLOS ONE’s publication criteria as it currently stands. Therefore, we invite you to submit a revised version of the manuscript that addresses the points raised during the review process.

Firstly, apologies for the long wait for this decision. We received comments from a statistical reviewer, who has requested more information on the KAMILA approach. I think the reviewer has provided a balanced argument, where they suggest that some more detail is provided on how this clustering method works, without having to recap everything, and also asks you to provide some justification on why this methodology was chosen. I hope that you find this feedback useful, and we look forward to receiving your revised manuscript. 

We look forward to receiving your revised manuscript.

Kind regards,

Hanna Landenmark

Staff Editor

PLOS ONE

hlandenmark@plos.org

Journal Requirements:

Reviewers' comments:

Reviewer's Responses to Questions

**Comments to the Author**

1. If the authors have adequately addressed your comments raised in a previous round of review and you feel that this manuscript is now acceptable for publication, you may indicate that here to bypass the “Comments to the Author” section, enter your conflict of interest statement in the “Confidential to Editor” section, and submit your "Accept" recommendation.

Reviewer #3: All comments have been addressed

Reviewer #4: (No Response)

2. Is the manuscript technically sound, and do the data support the conclusions?

Reviewer #3: Yes

Reviewer #4: Yes

3. Has the statistical analysis been performed appropriately and rigorously? 

Reviewer #3: Yes

Reviewer #4: I Don't Know

4. Have the authors made all data underlying the findings in their manuscript fully available?

Reviewer #3: Yes

Reviewer #4: Yes

5. Is the manuscript presented in an intelligible fashion and written in standard English?

Reviewer #3: Yes

Reviewer #4: Yes

6. Review Comments to the Author

Reviewer #3: Congratulations on an excellent work well done. This paper will serve as a guideline for many epidemiologists, not only in Europe but Worldwide

Reviewer #4: Major Revision

This is a very complex paper describing the application of clustering techniques to help identify groups of patients with COVID-19 who essentially then have differing risks of, for example, death from their disease.

Clustering techniques essentially take variables, ranging from the patient characteristics at presentation, to clinical and laboratory measures. In this study these appear to concern 292 such measures which are then reduced by some means (not fully explained) to 36. In brief, these remaining variables are then exposed to a so-called KAMILA clustering technique which appears (see Table 1 for example) to be reduced to 20 for the Wave 1 group of COVID patients. Sadly, no explanation of how KAMILA works is provided although this is perhaps not so surprising as the reference to Foss and Markatou (2018) describing the technique comprises 43 pages of complex text. Nevertheless, without some explanation of how KAMILA works the reader is left facing a totally ‘black box’ (a total mystery) procedure.

I presume KAMILA somehow reduces the number of variables (here from 36 to 20) and then creates several 20-dimensional clouds of these variables which if distinct (although perhaps overlapping) are then identified as the Clusters. However, whatever this process, the two groups (Wave 1: Cluster 1 & Cluster 2) identified for Table 3 appear to distinguish them by having higher In-hospital mortality, need for ICU assistance, and for Invasive mechanical assistance. Similar findings are reported for Waves 2, 3 and 4.

Personally, I am not very comfortable with the complex processes involved here, but more importantly any non (cluster) technical reader will be mystified by the presentation. I note that the authors all appear to be statisticians (as I am) but I strongly feel that the paper could be much improved with a simple review of the clustering process and some clinical input to reinforce the value of this approach.

Incidentally, this paper without Page and Line numbers included, made reviewing very difficult.

7. PLOS authors have the option to publish the peer review history of their article (what does this mean?). If published, this will include your full peer review and any attached files.

Reviewer #3: **Yes: **Mahmoud Elfiky

Reviewer #4: No

---

## [Decision Letter · Decision Letter 0]

14 Mar 2023

PONE-D-22-31704

COVID-19 patient profiles over four waves in Barcelona metropolitan area: A clustering approach

PLOS ONE

Dear Dr. Fernández,

Thank you for submitting your manuscript to PLOS ONE. After careful consideration, we feel that it has merit but does not fully meet PLOS ONE’s publication criteria as it currently stands. Therefore, we invite you to submit a revised version of the manuscript that addresses the points raised during the review process.

We look forward to receiving your revised manuscript.

Kind regards,

Sonu Bhaskar, MD PhD

Academic Editor

PLOS ONE

Journal Requirements:

a) Did participants provide their written or verbal informed consent to participate in this study?

3. Please amend your current ethics statement to include the full name of all ethics committees/institutional review boards that approved your specific study.

DF, NP, and GM have been supported by l’Agència de Gestió d'Ajuts Universitaris i de Recerca (AGAUR) de la Generalitat de Catalunya (Spain) [2020PANDE00148] (https://agaur.gencat.cat/en/inici/index.html). DF and NP have been supported by the Ministerio de Ciencia e Innovación (Spain) [PID2019-104830RB-I00/ DOI (AEI): 10.13039/501100011033] (https://www.aei.gob.es/en/announcements/announcements-finder/proyectos-idi-2019-modalidades-retos-investigacion-generacion). The funders had no role in study design, data collection and analysis, decision to publish, or preparation of the manuscript.

However, funding information should not appear in the Acknowledgments section or other areas of your manuscript. We will only publish funding information present in the Funding Statement section of the online submission form. 

DF, NP, and GM have been supported by l’Agència de Gestió d'Ajuts Universitaris i de Recerca (AGAUR) de la Generalitat de Catalunya (Spain) [2020PANDE00148] (https://agaur.gencat.cat/en/inici/index.html). DF and NP have been supported by the Ministerio de Ciencia e Innovación (Spain) [PID2019-104830RB-I00/ DOI (AEI): 10.13039/501100011033] (https://www.aei.gob.es/en/announcements/announcements-finder/proyectos-idi-2019-modalidades-retos-investigacion-generacion). The funders had no role in study design, data collection and analysis, decision to publish, or preparation of the manuscript.

6. One of the noted authors is a group or consortium DIVINE project. In addition to naming the author group, please list the individual authors and affiliations within this group in the acknowledgments section of your manuscript. Please also indicate clearly a lead author for this group along with a contact email address.’ 

7. Your ethics statement should only appear in the Methods section of your manuscript. If your ethics statement is written in any section besides the Methods, please move it to the Methods section and delete it from any other section. Please ensure that your ethics statement is included in your manuscript, as the ethics statement entered into the online submission form will not be published alongside your manuscript. 

Additional Editor Comments:

Thank you for submitting your work to PLOS One. Based on feedback from the reviewers, and careful review of your submission, I invite you to revise your manuscript and provide point by point rebuttal to comments provided by the reviewer/s. I look forward to reading revised version of your submission.

Reviewers' comments:

Reviewer's Responses to Questions

Comments to the Author

1. Is the manuscript technically sound, and do the data support the conclusions?

Reviewer #1: Partly

Reviewer #2: Yes

Reviewer #3: Yes

2. Has the statistical analysis been performed appropriately and rigorously? 

Reviewer #1: Yes

Reviewer #2: Yes

Reviewer #3: Yes

3. Have the authors made all data underlying the findings in their manuscript fully available?

Reviewer #1: Yes

Reviewer #2: Yes

Reviewer #3: Yes

4. Is the manuscript presented in an intelligible fashion and written in standard English?

Reviewer #1: No

Reviewer #2: Yes

Reviewer #3: Yes

5. Review Comments to the Author

Reviewer #1: In this article, the Authors report the results of a prospective multicentre cohort study based on patient clusterization during four COVID-19 waves within the southern Barcelona metropolitan area. Their aim is to identify different patient profiles and to relate them to the severity degrees of the outcomes.

The study was conducted appropriately, with a methodology employing a clustering classification based on multistep procedure. However, description of the steps undertaken was lacking, with grammatical and syntax errors that made the reading difficult to follow.

Furthermore, we recommend extensive English language editing for the whole document.

In the list below, Authors can find a number of points that specifically need to be addressed in order to improve the quality of the article.

Abstract

Objectives: syntax and meaning of the sentence are not clear.

Methods: the phrase “level of obesity” is not exact. In both table 1 and 2 it is only specified the number of patients with a BMI score above 30 and not the level of obesity. Use “number of obese patients”, instead.

Introduction

In the first sentence it is better to use “a.k.a” in the extended form “also known as”.

Methods

a) Setting and participants:

o Was the informed consent necessary? Did the Authors acquired it?

o The sentence ”All patients were adults (>18 years old) and admitted with PCR-proven SARS-CoV-2 infection for at least 48 hours” is not clear: did the PCR test has to become positive in the 48 hours before or following admission?

b) Analytic process:

1. Collecting and formatting of study data: in the second sentence it is not specified who did in particular fill the questionnaires. Did the Authors or other physicians do it?

2. Interpretation of cluster profiles:

− In the sentence “…we shared our analytic results with the physicians to define…” which are the physicians the Authors are referring to? How were the distinct profiles defined? Was there consensus between the MDs in the definitions?

− In the last sentence it is not clear what the Authors are trying to say.

Results

KAMILA Results:

o The sentence “Generally speaking, the more determined variables for the three first waves to find the clusters were patient’s age and level of obesity, the number of comorbidities, oxygen support requirement, and various severity scores as SaFi illness, which is defined as the ratio between the oxygen saturation and the fraction of inspired oxygen” is ambiguous, so it is necessary to review it.

o In the same sentence as above, it’s more appropriate the use of “i.e” instead of “a.k.a”.

Patient Profiles:

o When referring to age or ages use “lower” instead of “smaller”.

o In the last sentence of the first paragraph “degrees of obesity” is not correct (see Abstract in this review).

o The 2nd and 3rd paragraph need a significant language and syntax editing.

o In the 4th paragraph the sentence “Between the two non-vaccinated….combined with larger ages” has multiple issues:

Is better to use “less-vaccinated” than “non-vaccinated” because in cluster 1 and 2 the vaccination rate is low but not null.

The full regimen vaccination rate is actually lower in the first cluster (15.3%) and not in the 2nd (16.6 %).

“Larger ages” is not correct, better to use “a wider age’s range”.

o In the sentence “The elderly patients (cluster 3)….the survivors of previous waves.” it’s not clear how the Authors draw these conclusions. Have only the patients in Cluster 3 survived the previous COVID-19 waves? Did these patients catch SARS-Cov-2 during previous waves?

o eTable 3 should be added to the main article because it is useful to better understand the different outcomes in the different clusters.

Discussion

The meaning of the last sentence of the 1st paragraph, as highlighted before in the results, is not clear. Does it mean that the patients in the other clusters didn’t survive the first three waves?

In the 2nd paragraph the sentence “The reason is that….those outcomes of COVID-19” is not clear.

In the 2nd sentence of the 3rd paragraph the sentence “were evidenced that did not work” is not correct

Conclusions

The conclusion is brief. It would be nice if the Authors deepened their view about how the clustering approach could help the physicians in a practical setting. For instance, it would be interesting to have some actual examples on how the patient’s management should be differentiated according to the different clustering at hospital admission.

Reviewer #2: Dear Author/s

Greetings

The article does not present a scientific novelty. It seems only a brief assessment of the Covid 19 waves. Therefore, it is not suitable for publication in the journal.

Best regards

Reviewer #3: Excellent work in collecting and analysing the data.

Consider adding a comparison to other large metropolitan areas in Spain or Europe

Please add a better quality image of Figure 1. Bar plots of the % of the outcome

6. PLOS authors have the option to publish the peer review history of their article (what does this mean?). If published, this will include your full peer review and any attached files.

Do you want your identity to be public for this peer review?

 For information about this choice, including consent withdrawal, please see our Privacy Policy.

Reviewer #1: 

Yes: 

Andrea Orsi

Reviewer #2: 

Yes: 

Yavuz AYAR

Reviewer #3: 

Yes: 

Mahmoud Elfiky, MD, GCSRT

---

## [Author Response · Author response to Decision Letter 1]

22 Mar 2024

The response to reviewers documents has been included in the attached files.

---

## [Decision Letter · Decision Letter 2]

4 Apr 2024

COVID-19 patient profiles over four waves in Barcelona metropolitan area: A clustering approach

PONE-D-22-31704R2

Dear Dr. Fernandez,

We’re pleased to inform you that your manuscript has been judged scientifically suitable for publication and will be formally accepted for publication once it meets all outstanding technical requirements.

Kind regards,

Dong Wook Chang

Academic Editor

PLOS ONE

Additional Editor Comments (optional):

Thank you for your revisions and additions to this very interesting manuscript. It is technically very complex, and I believe the additional information KAMILA and the tables have significantly improved the readability of this work.

Reviewers' comments:

Reviewer's Responses to Questions

**Comments to the Author**

1. If the authors have adequately addressed your comments raised in a previous round of review and you feel that this manuscript is now acceptable for publication, you may indicate that here to bypass the “Comments to the Author” section, enter your conflict of interest statement in the “Confidential to Editor” section, and submit your "Accept" recommendation.

Reviewer #3: All comments have been addressed

Reviewer #4: All comments have been addressed

2. Is the manuscript technically sound, and do the data support the conclusions?

Reviewer #3: Yes

Reviewer #4: Yes

3. Has the statistical analysis been performed appropriately and rigorously? 

Reviewer #3: Yes

Reviewer #4: Yes

4. Have the authors made all data underlying the findings in their manuscript fully available?

Reviewer #3: Yes

Reviewer #4: Yes

5. Is the manuscript presented in an intelligible fashion and written in standard English?

Reviewer #3: Yes

Reviewer #4: Yes

6. Review Comments to the Author

Reviewer #3: Excellent work as stated before. I praise your efforts in detailing your Statistical Analysis eloquently and carefully

Reviewer #4: Accept

The authors have made substantial changes to the paper which, I hope, will make it easier for a clinical reader to digest. I have no further comments.

7. PLOS authors have the option to publish the peer review history of their article (what does this mean?). If published, this will include your full peer review and any attached files.

Reviewer #3: **Yes: **Mahmoud Elfiky

Reviewer #4: No

---

## [Editor Report · Acceptance letter]

24 Apr 2024

PONE-D-22-31704R2 

PLOS ONE

Dear Dr. Fernández, 

I'm pleased to inform you that your manuscript has been deemed suitable for publication in PLOS ONE. Congratulations! Your manuscript is now being handed over to our production team.

Kind regards, 

on behalf of

Dr. Dong Wook Chang 

Academic Editor

PLOS ONE